# Vitamin D and Cardiovascular Disease: An Updated Narrative Review

**DOI:** 10.3390/ijms22062896

**Published:** 2021-03-12

**Authors:** Armin Zittermann, Christian Trummer, Verena Theiler-Schwetz, Elisabeth Lerchbaum, Winfried März, Stefan Pilz

**Affiliations:** 1Clinic for Thoracic and Cardiovascular Surgery, Heart and Diabetes Center North Rhine-Westphalia (NRW), Ruhr University Bochum, 32545 Bad Oeynhausen, Germany; azittermann@hdz-nrw.de; 2Department of Internal Medicine, Division of Endocrinology and Diabetology, Medical University of Graz, 8036 Graz, Austria; christian.trummer@medunigraz.at (C.T.); verena.schwetz@medunigraz.at (V.T.-S.); elisabeth.lerchbaum@medunigraz.at (E.L.); 3Clinical Institute of Medical and Chemical Laboratory Diagnostics, Medical University of Graz, 8036 Graz, Austria; winfried.maerz@synlab.de; 4SYNLAB Holding, Deutschland GmbH, 68159 Mannheim and Augsburg, Germany; 5V^th^ Department of Medicine (Nephrology, Hypertensiology, Rheumatology, Endocrinology, Diabetology, Lipidology), Medical Faculty Mannheim, University of Heidelberg, 68167 Mannheim, Germany

**Keywords:** vitamin D, cardiovascular, heart, atherosclerosis, epidemiology, vitamin D receptor, mortality, chronic kidney disease, calcium

## Abstract

During the last two decades, the potential impact of vitamin D on the risk of cardiovascular disease (CVD) has been rigorously studied. Data regarding the effect of vitamin D on CVD risk are puzzling: observational data indicate an inverse nonlinear association between vitamin D status and CVD events, with the highest CVD risk at severe vitamin D deficiency; however, preclinical data and randomized controlled trials (RCTs) show several beneficial effects of vitamin D on the surrogate parameters of vascular and cardiac function. By contrast, Mendelian randomization studies and large RCTs in the general population and in patients with chronic kidney disease, a high-risk group for CVD events, largely report no significant beneficial effect of vitamin D treatment on CVD events. In patients with rickets and osteomalacia, cardiovascular complications are infrequently reported, except for an increased risk of heart failure. In conclusion, there is no strong evidence for beneficial vitamin D effects on CVD risk, either in the general population or in high-risk groups. Whether some subgroups such as individuals with severe vitamin D deficiency or a combination of low vitamin D status with specific gene variants and/or certain nutrition/lifestyle factors would benefit from vitamin D (metabolite) administration, remains to be studied.

## 1. Introduction

It has been known for about 100 years that vitamin D can prevent and cure bone diseases such as rickets and osteomalacia. However, it has also been known for a similar time period that supraphysiological doses of vitamin D can cause vascular calcification [1]. Nevertheless, more recent ecological and observational data also suggest that an insufficient or deficient vitamin D status may increase the risk of cardiovascular diseases (CVD) [2]. CVD is a leading cause of morbidity and mortality globally, and an inadequate vitamin D status is a worldwide issue as well; therefore, the role of vitamin D for preventing CVD outcomes has been rigorously studied during the last two decades [3,4].

In this narrative review, we will provide an updated overview of contemporary preclinical data along with epidemiological and experimental human evidence regarding vitamin D and CVD.

## 2. Vitamin D Metabolism

Vitamin D metabolism is unique (Figure 1) because it can be orally ingested but also synthesized in the human skin by solar ultraviolet B radiation. Dietary intake from non-fortified foods is usually low. Although skin synthesis is very effective, it is also low or even negligible during the winter half-year at geographical latitudes 35° N (or south) and beyond [5]. Both dietary and lifestyle factors contribute to the high prevalence of inadequate vitamin D status globally (see below). Adipose tissue and skeletal muscle are the major storage sites for vitamin D [6].

Vitamin D metabolism includes synthesis from 7-dehydrocholesterol, its sequential hydroxylation by the mitochondrial cytochrome P450 enzymes 25-hydroyxlase (gene CYP2R1) and other 25-hydroxylases in the liver, and 1-α hydroxylase (gene CYP27B1) in the kidney into the active metabolite 1,25-dihydroxyvitamin D (1,25(OH)_2_D). In the circulation, 25-hydroxyvitamin D (25(OH)D) and 1,25(OH)_2_D are mainly bound to vitamin D binding protein (DBP) and, to a lesser extent, albumin. Usually, only a small fraction of unbound vitamin D metabolites are biologically active, with the exception of some tissues such as kidney cells which are also capable of internalizing DBP-bound 25(OH)D by a megalin–cubilin-mediated process. The half-life of 25(OH)D in the circulation (10–20 days) is much higher than the half-life of 1,25(OH)_2_D (10–20 h), due to the higher affinity for DBP of the former compared with the active hormone [7]. Circulating 1,25(OH)_2_D is tightly regulated by parathyroid hormone (PTH) and fibroblast growth factor 23 (FGF-23) to maintain plasma calcium and phosphate within their relatively narrow physiological ranges. FGF-23 inhibits 1α-hydroxylase and activates 24-hydroxylase (gene CYP24A1), resulting in decreased 1,25(OH)_2_D and increased 24,25(OH)_2_D levels [8], whereas 1α-hydroxylase is activated by PTH as a consequence of low plasma calcium levels [9]. Circulating 1,25(OH)_2_D also depends on substrate availability and is reduced at low 25(OH)D concentrations, i.e., <25 nmol/L [9]. CYP24A1 is not only induced by FGF-23, but also by 1,25(OH)_2_D and 25(OH)D and is one of the most highly inducible genes in humans, which is likely to be a critical factor in the relatively large therapeutic window of vitamin D [10]. We wish to emphasize that this brief description of vitamin D metabolism is only a very simplified model that does not cover the full complexity of the regulation of vitamin D metabolism, extra-hepatic and extra-renal vitamin D metabolizing enzymes, data on long-term tissue storage of vitamin D metabolites, or the potential biologically role of native vitamin D (and its isoforms) itself, for which we refer the reader to more in-depth publications covering these topics [11,12,13].

The compound 1,25(OH)_2_D is a steroid hormone (Figure 2). It exerts rapid non-genomic cellular effects via membrane-bound receptors and, as the main pathway, genomic cellular effects via cytosolic receptors. Binding of 1,25(OH)_2_D induces a conformational change in the vitamin D receptor (VDR), leading to hetero-dimerization with the retinoid X receptor and ultimately to translocation of this complex into the nucleus, where it binds to vitamin D response elements in the promoter region of target genes. A variety of extra-renal cells also exhibit 1α-hydroxylase activity, indicating that in these cells the biological effects are determined by the sum of circulating and locally produced 1,25(OH)_2_D [14].

Polymorphisms of the genes encoding for proteins influencing circulating 25(OH)D concentrations such as 7-dehydrocholesterol reductase, DBP, 25-hydroxylase, and 24-hydroxylase have been described. Moreover, several gene variants of the VDR are known (see below).

Circulating 25(OH)D is generally accepted as the integral marker of human vitamin D status. Based on circulating 25(OH)D, there are different definitions of an adequate vitamin D status. Many nutrition societies define vitamin D sufficiency as a 25(OH)D level of more than 50 nmol/L [15,16,17,18], whereas the Endocrine Society’s definition is a level more than 75 nmol/L [19]. Circulating 25(OH)D concentrations <30 nmol/L or <25 nmol/L are often considered as the threshold for vitamin D deficiency [15,20]. Recommendations for daily vitamin D intake in the case of low or absent skin synthesis of vitamin D generally vary between 600 international units (IU) and 2000 IU [15,16,17,18]. Globally, inadequate vitamin D status is highly prevalent. Briefly, circulating 25(OH)D greater than 50 nmol/L is present in less than 50% of the world population, at least during the winter season [21]. In Europe, the prevalence of circulating 25(OH)D <50 nmol/L and <30 nmol/L is 40.4% and 13.0%, respectively [22]. In several low- to middle-income countries the prevalence of circulating 25(OH)D concentrations <25–30 nmol/L is between 40 and 90% in some groups of the general population, such as children, women, and older adults [23].

## 3. Vitamin D Effects on The Cardiovascular System

In general, vitamin D effects on cardiovascular health may be mediated either by effects on classic and emerging cardiovascular risk factors or by direct effects on the cardiovascular system [14,24,25]. While potential effects on certain cardiovascular risk factors have been specifically addressed by various other reviews, we focus on direct cardiovascular effects of vitamin D [25]. Some excellent reviews have summarized vascular and cardiac vitamin D effects on the cellular level [14,24]. In brief, both the VDR and the 1α-hydroxylase are present in vascular tissues such as endothelial cells and vascular smooth muscle cells (VSMCs), and also in cardiomyocytes [26]. In the vascular wall, 1,25(OH)_2_D has several beneficial genomic effects, including a reduction in thrombogenicity, a decrease in vasoconstrictors, an inhibition of oxidative stress and atherogenesis, an improvement of endothelial repair, a reduction in foam cell formation, and vascular relaxation and dilatation. However, 1,25(OH)_2_D can also induce the trans-differentiation of VSMCs into osteoblast-like cells, which may lead to vascular calcification [25,26]. In cardiomyocytes, 1,25(OH)_2_D can induce several genomic and non-genomic effects, which regulate intracellular calcium metabolism [27]. There is evidence that, similar to the regulation of circulating 1,25(OH)_2_D, the synthesis of 1,25(OH)_2_D in the heart and the vasculature is regulated by PTH and FGF-23 [28,29]. In cardiomyocytes, FGF-23 results in hypertrophic growth [30]. Although vitamin D signaling in cardiomyocytes and the vascular wall is not completely clarified in detail, the available data indicate that 1,25(OH)_2_D plays a pivotal role for adequate cardiac and vascular function. Various effects of vitamin D on the cardiovascular system may, in addition to VDR activation, also be mediated by PTH (Figure 3).

## 4. Preclinical Data on Vitamin D and Cardiovascular Diseases

During the last two decades, several studies have investigated the effect of VDR ablation on the cardiovascular system in experimental animals. The main results have been summarized elsewhere [9]. Briefly, VDR deletion results in elevated production of renin and angiotensin II, leading to hypertension and cardiac hypertrophy [31,32]. Treatment of VDR knockout mice with the ACE inhibitor captopril reduces cardiac hypertrophy and normalizes atrial natriuretic peptide expression [32]. Cardiomyocyte-specific deletion of the VDR also results in cardiac hypertrophy, and treatment of neonatal cardiomyocytes with 1,25(OH)_2_D is partially able to suppress hypertrophy [33]. Moreover, vitamin D deficiency stimulates renin expression in normal mice, whereas the injection of 1,25(OH)_2_D reduces renin synthesis [34]. Similar to vitamin D excess, deletion of the VDR as well as diets low in vitamin D content stimulate osteoblast-like cell formation of vascular smooth muscle cells and aortic calcification [35,36]. VDR knockout mice also display increased thrombogenicity [37]. With respect to hypertension and the renin–angiotensin–aldosterone system (RAAS) activation, it has, however, been criticized that in VDR-null mice fed a normal diet, the changes may be due to secondary hyperparathyroidism and that VDR knock-out mice fed the so-called “rescue diet” enriched with calcium, phosphate, and lactose, which normalizes mineral homeostasis in these mice, did not show increased mean arterial pressure or RAAS activation [14]. Nevertheless, others have reported that in 1α-hydroxylase knockout mice, the protective role of 1,25(OH)_2_D on the development of hypertension, cardiac hypertrophy and impaired cardiac function along with an up-regulation of the RAAS in the renal and cardiac tissue was independent of plasma calcium and phosphate levels [38]. Although there is accumulating evidence from cell culture and animal studies on the involvement of VDR signaling in cardiovascular health, we have to be very cautious when translating these data into human pathophysiology, and should rather base our conclusions on clinical studies.

## 5. Data on Vitamin D and Cardiovascular Diseases in Humans

In humans, several types of studies may shed light on the effect of vitamin D on the cardiovascular system, including data from patients with rickets and osteomalacia, observational studies, genetic studies, and randomized controlled trials (RCTs).

### 5.1. Cardiovascular Disease in Severe Vitamin D Deficiency Such as Rickets and Osteomalcia

A number of case reports have been published associating nutritional rickets with heart failure, but not with other forms of CVD such as hypertension, stroke, or myocardial infarction. In all these cases, young patients also suffered from secondary hyperparathyroidism and hypocalcemia. The cardiac effects were effectively cured by vitamin D and calcium administration [39,40]. However, heart failure or other forms of CVD have not been described in genetic forms of rickets such as vitamin D-dependent rickets type I and II. They are based on a lack of the 1-alpha hydroxylase enzyme or a deletion of the VDR, respectively. Both genetic disorders can be caused by more than 60 different mutations [41]. Their clinical symptoms usually present in infancy or early childhood and are similar to the phenotype of nutritional vitamin D-deficient rickets, including secondary hyperparathyroidism and hypocalcemia. Probably, CVD has not been described in these disorders because they are both rare pediatric diseases with relatively few known cases.

Hypocalcemia with heart failure has indeed been reported in a 41-year-old female patient with nutritional osteomalacia. Her cardiac failure improved promptly on correcting the hypocalcemia by vitamin D and calcium [42]. Again, however, CVD has not been reported in cases with other forms of osteomalacia such as tumor-induced osteomalacia. This is a rare endocrine disorder of disturbed vitamin D metabolism, predominantly in middle-aged adults. It is characterized by hypophosphatemia, phosphaturia, and inappropriately low serum levels of 1,25(OH)_2_D. The biochemical changes are related to elevated FGF-23 concentrations [43,44,45].

Altogether, some toddlers or younger adults with rickets or osteomalacia may develop hypocalcemia-associated heart failure, but obviously not other forms of CVD. Conversely, significantly lower circulating vitamin D metabolites and plasma calcium concentrations have been reported in patients with advanced heart failure compared with healthy controls, with the lowest circulating 1,25(OH)_2_D concentrations and plasma calcium concentrations in the hypocalcemia range (<2.1 mmol/L) in those patients with early-onset of the disease [27].

### 5.2. Cardiovascular Events in General Populations According to Vitamin D Status

Numerous meta-analyses of observational studies have investigated the association of circulating 25(OH)D with CVD risk (Table 1).

The meta-analyses were based on up to about 180,000 individuals [46] and several thousand events. Data of these meta-analyses indicate that low vitamin D status is associated with a significantly increased risk of hypertension, cardiovascular events, and cardiovascular mortality. Some of these meta-analyses [46,47,48,49] not only compared low versus high 25(OH)D categories, but performed a dose–response analysis according to circulating 25(OHD concentrations. Data indicate a nonlinear increase in CVD risk at circulating 25(OH)D concentrations below 50 nmol/L compared with a circulating 25(OH)D concentration around 75 nmol/L, with the highest CVD risk at concentrations below 25 nmol/L. Hence, it can be concluded that vitamin D deficiency is a risk marker for adverse cardiovascular outcomes, but whether this reflects a causal association cannot be answered by observational data that are prone to confounding and reverse causation, i.e., that underlying diseases contribute to vitamin D deficiency.

### 5.3. Cardiovascular Disease According to Genetic Analyses of Vitamin D Metabolism

Genetic studies have the advantage that they are unaffected by lifestyle factors. With respect to vitamin D and CVD, genetic studies can be divided into two groups. Mendelian randomization is an analytical method that uses genetic variants as instrumental variables for modifiable risk factors such as circulating 25(OH)D that potentially affect CVD risk. Another group of genetic studies are association studies that analyze gene variants of non-modifiable risk factors such as VDR polymorphisms, which may affect CVD risk by influencing vitamin D signaling pathways.

Mendelian randomization studies reflect lifelong differences in vitamin D status. They are less susceptible to confounding and reverse causality bias than observational studies. Therefore, these studies are located at the interface between traditional observational epidemiology and interventional trials. About 7–8 genes with single nucleotide polymorphisms (SNPs) have been identified which affect circulating 25(OH)D [56] by influencing the synthesis, transport or metabolism of 25(OHD). A few publications using the Mendelian randomization approach have investigated the association of genetically determined 25(OH)D with CVD outcomes such as hypertension, stroke, coronary artery disease, myocardial infarction, and cardiovascular mortality [57,58,59,60,61]. The aforementioned studies used one [60] to four [59] SNPs to perform the analyses. Four publications reported no significant relationship on CVD risks such as ischemic stroke, ischemic heart disease, coronary artery disease, myocardial infarction, and CVD mortality [57,58,59,60]. In one study, each 10% increase in genetically determined 25(OH)D concentration was associated with a significant blood pressure reduction of 0.3 mm Hg and a significant reduction in the risk of hypertension [61]. Thus, taken together, an effect of genetically determined vitamin D status on clinically relevant CVD events was absent, or at best, modest. Nevertheless, one major limitation of Mendelian randomization studies is that the SNPs affecting circulating 25(OH)D account for only about 7.5% of variations in circulating 25(OH)D, an effect that is much lower than the effect achieved by oral vitamin D supplements or skin synthesis of vitamin D. Therefore, Mendelian randomization studies on vitamin D and CVD are subject to weak instrument bias [62]. Moreover, because the study participants of the aforementioned investigations were adults of different ages, results of many individuals might reflect the association between 25(OH)D and disease progression, rather than disease occurrence.

Several meta-analyses have also investigated the association of the Apa 1, Fok 1, Taq 1, and Bsm 1 polymorphisms of the VDR gene or polymorphisms of the CYP24A1 gene with CVD risk [63,64,65,66,67,68]. Some of the meta-analyses reported significant associations of VDR Bsm 1, Apa 1, and Fok 1 gene variants with CVD parameters such as hypertension, vascular and microvascular complications, or coronary artery disease [63,64,65,66], whereas another meta-analysis reported no associations of these gene variants with coronary artery disease [67]. One investigation reported that a common variant in the CYP24A1 gene was associated with coronary artery calcification in three independent populations [68]. However, none of the aforementioned meta-analyses reported *p*-values for the described associations of <10^−7^, which is considered to be the threshold for reliable associations of a gene polymorphisms with a disease outcome.

### 5.4. Cardiovascular Surrogate Parameters in Vitamin D Supplemenation Trials

There is general agreement that convincing scientific evidence can at best be achieved by adequately powered RCTs. This means that the vitamin D supplement used should be able to beneficially influence surrogate parameters of CVD or to prevent CVD events. To prove causality, CVD should be the primary outcome variable.

#### 5.4.1. Vascular Parameters

In the vasculature, the interactions among endothelial cells, VSMCs, adventitial tissues, and inflammatory cells are substantially involved in vascular health and disease. Vascular function can be assessed by measuring aortic pulse wave velocity (PWV, the velocity at which the blood pressure pulse propagates through the circulatory system) and augmentation index (AI, the ratio of late systolic pressure to early systolic pressure). Both parameters are accepted measures of arterial stiffness. Moreover, ultrasonographic measurement of flow-mediated dilation (FMD, dilation of an artery when blood flow increases in that artery following a transient period of forearm ischemia) can be used to assess endothelial function. Several meta-analyses were unable to show any significant effects of vitamin D supplementation on PWV, AI, or FMD [69,70,71,72] (Table 2).

A large meta-analysis [73] used trial-level meta-analyses as well as individual-participant data (IPD) meta-analyses. Vitamin D3 daily dose equivalents ranged from 900 to 5700 IU, and study duration was four weeks to 12 months. Again, no consistent benefit of vitamin D was observed on PWV, AI, or FMD, either in trial-level meta-analyses, or in IPD-level meta-analyses. Moreover, vitamin D did not improve parameters of vascular function in subgroups with baseline 25(OH)D concentrations <25 nmol/L. However, trial-level meta-regression analysis indicated a slightly greater treatment effect for FMD by higher vitamin D dose. Similarly, another meta-analysis [74] reported that a sub-analysis and meta-regression revealed a tendency for AI and FMD to increase as weekly vitamin doses increased. Likewise, another meta-analysis [75] reported that a daily dose of vitamin D3 ≥2000 IU significantly improved PWV. Notably, this daily dose is higher than the daily dose of 600 to 800 IU that is recommended by several nutrition societies [15,16,17,18].

#### 5.4.2. Cardiac Parameters

Echocardiographic measurements of left ventricular ejection fraction (LVEF) and left ventricular end-diastolic diameter (LVEDD) are frequently used tools to assess cardiac function and the risk of heart failure. In this regard, available meta-analyses showed inconsistent results with no significant improvement by vitamin D supplementation on LVEF and LVEDD in two analyses [76,77], and a significant dose-dependent increase in LVEF and a decrease in LVEDD in another analysis, suggesting an inhibition of ventricular remodeling [78] (Table 2). Again, daily doses were substantially higher than the recommended daily dose of 600 to 800 IU/d (Table 2). None of the meta-analyses investigated the effect of vitamin D on cardiac function in subgroups with initially insufficient or deficient vitamin D status. Subgroup analysis in one of the aforementioned meta-analyses documented no significant improvement [77], but did reveal a significant benefit on LVEF by vitamin D supplementation if the analysis was restricted to the subgroup of study participants without calcium co-administration. Notably, the Atherosclerosis Risk in Communities (ARIC) study [79] has already reported a non-linear association of plasma calcium with the risk of incident heart failure, with the lowest risk at calcium levels of 2.25 mmol/L and a progressive increase up to 2.75 mmol/L. In line with these findings, vitamin D supplementation improved LVEF in heart failure patients with baseline plasma calcium of 2.4 nmol/L only in the age group in which in-study plasma calcium remained constant (≥50 years), but not in the age group with a significant vitamin D-associated in-study increase in plasma calcium (<50 years) [80], indicating that plasma calcium may be crucial for the vitamin D effect in heart failure (see below). In the meta-analysis by Zhao et al. [78], a subgroup analysis revealed that vitamin D supplementation was more effective in reducing the LVEDD and increasing the LVEF in patients with reduced ejection fraction than in patients without reduced ejection fraction. Heart failure with preserved LVEF accounts for about 40–50% of incident heart failure cases overall [81]; therefore, a beneficial vitamin D effect on the cardiac system, if any, would thus be limited in a substantial subgroup of patients.

#### 5.4.3. Lipid Parameters

Dyslipoproteinemia is a well-known risk factor for CVD. Notably, the precursor of vitamin D, 7-dehydrocholesterol, can not only be converted by solar UVB radiation to pre-vitamin D3, but alternatively also by an enzymatic reaction, mediated by the enzyme 7-dehydrocholesterol-reductase, to cholesterol. A large meta-analysis has summarized data from RCTs regarding the effect of vitamin D supplementation on lipid parameters [71]. The average vitamin D dose was 3000 IU/day; two-thirds of the studies had mean baseline 25(OH)D levels <50 nmol/L, and the increase in circulating 25(OH)D was 48 ± 23 nmol/L. Data indicate a small, but significant, reduction in total cholesterol, LDL-cholesterol, and triglycerides of −0.15 (95%CI:−0.25 to −0.04) mmol/L, −0.10 (95%CI:−0.20 to −0.003) mmol/L, and −0.12 (95%CI:−0.23 to −0.003) mmol/L, respectively, and an increase in HDL-cholesterol of 0.09 (95%CI: 0.00 to 0.17) mmol/L by vitamin D supplementation. There was substantial heterogeneity among studies, and subgroup analyses indicate that the vitamin D effects on triglycerides and HDL-cholesterol were greater if study participants were supplemented for ≥6 months. However, there was no significant vitamin D effect according to baseline 25(OH)D concentration, daily vitamin D dose, or calcium co-administration on lipid parameters. Similar effects of vitamin D supplementation were reported in a more recent meta-analysis [82] for total cholesterol, LDL-cholesterol and triglycerides. In contrast to the aforementioned earlier meta-analysis, a non-significant decrease of −0.10 (95%CI: −0.28 to 0.09) mmol/L was reported for HDL-cholesterol. In that meta-analysis, the improvements in total cholesterol and triglycerides were more pronounced in participants with baseline 25(OH)D below 50 nmol/L [82]. Apart from this, there is also evidence from a proof of principle study by Greco et al. that vitamin D supplementation may modulate serum lipoprotein functions related to macrophage cholesterol homeostasis [83].

## 6. Cardiovascular Events in Large Vitamin D Supplementation Trials

The results of three large randomized controlled vitamin D supplementation trials on CVD outcomes as primary endpoints have been published [84,85,86]. Briefly, the ViDA (Vitamin D Assessment) trial [84] included 5110 male and female New Zealanders, aged 50 to 84 years, who received either a bolus dose of 100,000 IU vitamin D3 or a placebo monthly for a median follow-up of 3.3 years. Vital (Vitamin D And Omega-3 Trial) [85] randomized 25,871 U.S. men aged ≥50 and women aged ≥55 years and used a two-by-two factorial design to randomize study participants to a vitamin D3 dose of 2000 IU per day, and a dose of 1 g omega-3 fatty acids per day. Median follow-up was 5.3 years. DO-Health (Vitamin D3–Omega-3–Home Exercise–HeALTHy Ageing and Longevity Trial) [86] randomized 2157 European adults from different countries aged 70 years or older for three years of intervention in one of the following groups: 2000 IU/d of vitamin D3, 1 g/d of omega-3 fatty acids, or a strength-training exercise program. In all three RCTs, mean baseline circulating 25-hydroxyvitamin D concentrations were above 50 nmol/L, and mean in-study 25(OH)D concentrations increased to at least 94 nmol/L.

In ViDA [84], the incidence of the primary endpoint (fatal and non-fatal CVD events) was comparable between the vitamin D group (11.8%) and the placebo group (11.5%). Similar results were seen in that trial for secondary endpoints such as myocardial infarction, angina pectoris, heart failure, hypertension, arrhythmias, arteriosclerosis, stroke, and venous thrombosis. Likewise, in VITAL [85], neither the primary endpoint (composite of myocardial infarction, stroke, and cardiovascular mortality) differed significantly between the vitamin D group (*n* = 396 events) and the placebo group (*n* = 409 events), nor an expanded composite of major cardiovascular events plus coronary revascularization (vitamin D group: *n* = 536 events; placebo group: *n* = 558 events). In DO-Health [86], two of the six primary endpoints were CVD endpoints (systolic and diastolic blood pressure). There were no statistically significant benefits of vitamin D supplementation on either of these two parameters. In none of the three RCTs did results change substantially when analyses were restricted to study participants with baseline 25(OH)D concentrations <50 nmol/L. Notably, none of the trials explicitly reported the prevalence of baseline 25(OH)D concentrations <30 nmol/L, but these numbers were apparently very low.

Altogether, all three trials documented similar results with respect to vitamin D and CVD-related parameters, independent of frequency and dosing of vitamin D administration. The results of these three large trials, in which CVD parameters were the primary outcome parameters, are in general agreement with earlier meta-analyses of RCTs [87,88,89,90,91] and one umbrella review of RCTs [92], which were primarily based on studies in which CVD outcomes were only secondary endpoints. All these earlier systematic reviews also reported no beneficial effects of vitamin D supplementation on CVD outcomes, but do, at least, support the cardiovascular safety of vitamin D supplements at relatively high doses.

## 7. Cardiovascular Events in Patients with Chronic Kidney Disease (CKD)

In the clinical setting, chronic kidney disease is an illness of particular interest with respect to vitamin D and CVD: primarily, because both low circulating 25(OH)D and low 1,25(OH)_2_D concentrations are frequent findings in CKD patients [93,94]; secondly, because the risk of vascular calcification and CVD is substantially higher in patients with CKD than in individuals with preserved kidney function [95]; and thirdly, because impaired kidney function is considered to contribute to the loss of homeostatic control of serum calcium concentrations and may thus influence the cut-off point defining the toxicity of vitamin D and calcium [15]. There is evidence that vitamin D administration of >2000 IU/d vitamin D or activated vitamin D significantly increased concentrations of the cardiovascular risk marker FGF-23, especially in patients with end-stage kidney/heart failure [96]. An ancillary study to the VITAL trial in 1312 participants with diabetes mellitus at baseline showed no difference in eGFR changes or albuminuria over five years of treatment with vitamin D compared with placebo [97]. These data support the assumption that long-term supplementation of 2000 IU is safe with respect to kidney function. A re-analysis of the extended-release calcifediol trials in CKD patients that used doses of 25(OH)D resulting in very high circulating 25(OH)D concentrations found no significant differences in biochemical safety parameters such as plasma calcium, phosphate, FGF-23, and eGFR, even up to mean circulating 25(OH)D concentrations of 235 nmol/L [98]. From the available data, it is, however, not completely clear whether the use of vitamin D supplements achieving high circulating 25(OH)D concentrations are safe in CKD patients.

Regarding vitamin D administration in patients with CKD, the necessity to perform RCTs in addition to observational studies is emphasized by a meta-analysis, which summarized the data of observational studies and RCTs of vitamin D administration on cardiovascular mortality [99]. In the observational studies (21 studies, 221,610 patients), bolus administration of high vitamin D doses or the administration of activated vitamin D was associated with a reduced risk ratio (RR) for cardiovascular mortality (relative risk 0.55; 95%CI 0.34 to 0.77). In the meta-analysis of RCTs (17 studies, 1819 patients), however, administration of vitamin D or activated vitamin D did not significantly reduce cardiovascular mortality (RR = 0.93; 95%CI 0.36–2.44). In a relatively large trial [100] in patients undergoing maintenance hemodialysis (976 patients), the risk of a composite measure of fatal and nonfatal cardiovascular events even tended to be higher in patients receiving activated vitamin D versus usual care, especially when the per protocol set was analyzed (adjusted hazard ratio 1.36; 95%CI: 0.99–1.87; *p* = 0.06).

Altogether, data from RCTs do not support the recommendation of vitamin D supplement use in the general, older population to prevent cardiovascular events. Moreover, neither the administration of native vitamin D nor activated vitamin D show clear benefits in patients with CKD. It is, however, also noteworthy that neither the RCTs in the general population nor the RCTs in patients with CKD reported significant harms on CVD outcomes, although some concerns in CKD patients remain.

## 8. Discussion of the Available Evidence

There are several lines of evidence that can be used to assess the association of vitamin D with CVD. Despite accumulating preclinical and clinical data, the association between vitamin D and CVD remains puzzling. Although preclinical data indicate several beneficial effects of vitamin D on the cardiac and vascular system, results from genetic studies and RCTs demonstrate, at best, a modest effect of vitamin D on CVD surrogate parameters, but not on CVD events. Some beneficial results on surrogate parameters of vascular and cardiac health were obtained with much higher doses than recommended to prevent vitamin D deficiency [15,16,17,18]. Several reasons may be responsible for the inconsistent results. Firstly, animal models may only partly be able to mirror the situation in humans. In humans, CVD is usually of multifactorial origin and not caused by a single gene defect. Secondly, most RCTs included a high percentage of patients who were initially not vitamin D-deficient. In line with this suggestion, results from observational studies indicate the strongest association of vitamin D status with various CVD outcomes at circulating 25(OH)D <25 nmol/L. However, we should not be too enthusiastic that the restriction of RCTs to vitamin D-deficient patients would change study outcomes substantially. Unexplained confounding may have influenced the study results of observational studies. Even in patients with rickets and osteomalacia, CVD events have rarely been described in the medical literature as a problem of these diseases. The prevalence of rickets and osteomalacia was very high during the last century and is still high in some countries, therefore it should have been recognized if vitamin D deficiency was a major causal risk factor for CVD. In addition, some studies report the Mendelian randomization estimate in the context of traditional observational epidemiology. The observational data report a significantly higher risk of CVD mortality by a 20 nmol/L decrease in circulating 25(OH)D, whereas a genetically determined decrease of 25(OH)D by 20 nmol/L tended to decrease the risk of CVD mortality (relative risk 1.13, 95%CI:1.03–1.24 versus 0.77, 95%CI: 0.55–1.08) [58]. Therefore, the Mendelian randomization estimate provides strong evidence that the traditional observational estimate arises from confounding and/or reverse causality.

A third issue is that 1,25(OH)_2_D is the active hormone, which is usually tightly regulated. One may argue that gene variants influencing circulating 25(OH)D or vitamin D supplements increasing circulating 25(OH)D may not affect 1,25(OH)_2_D availability over a wide range of circulating 25(OH)D concentrations. However, even this argumentation seems to fall short, because some rare disorders with substantially reduced 1,25(OH)_2_D concentrations or absent vitamin D action obviously do not suffer from CVD events. Even in patients with low circulating 1,25(OH)_2_D and high CVD risk such as CKD patients, the administration of activated vitamin D obviously does not reduce CVD events. A novel RCT in hemodialysis patients does not even exclude the possibility that the administration of activated vitamin D may increase the risk of CVD events in those patients adhering to the study treatment. One can therefore speculate that low circulating 1,25(OH)_2_D concentrations are probably not causally related to the high prevalence of CVD events in these patients, although may be the result of FGF-23-mediated suppression of the 1α-hydroxylase, which may prevent the body from (calcium) and phosphate intoxication and thus from further vascular calcification.

Overall, the data are best for an association between vitamin D and heart failure. This assumption is supported by data from patients with nutritional rickets and osteomalacia, and also from meta-analyses of RCTs on vitamin D and cardiac function. Results have also repeatedly been supported by preclinical data in experimental animals. However, additional risk factors may influence heart failure incidence. In humans, the vitamin D effects in heart failure patients are probably related to plasma calcium levels, with detrimental effects of hypocalcemia as well as high normal calcium concentrations. Thus, the window for beneficial vitamin D effects on plasma calcium and on cardiac function may be narrow, if any, because adverse effects of vitamin D supplementation in association with increased plasma calcium concentrations have also been reported in patients with advanced heart failure [101]. Moreover, there are other factors that can slightly, but significantly, increase plasma calcium concentrations such as calcium supplementation [102] and osteoporosis-induced calcium resorption from bone [103], whereas physical activity reduces plasma calcium [104]. The effect of these factors on incidental heart failure and their interaction with vitamin D is largely unknown and may contribute to the null effect of vitamin D supplementation on heart failure risk in some large studies, i.e., the ViDA trial [84].

When summarizing data on vitamin D and CVD in this paper, we have to acknowledge that we did not discuss in detail the evidence for vitamin D, and common as well as emerging cardiovascular risk factors such as diabetes mellitus or inflammation, that may be relevant in mediating a potential link between vitamin D and CVD [105,106,107,108]. We are well aware that there is accumulating evidence on a role of vitamin D in, e.g., inflammation, but this did obviously not translate into an improved cardiovascular outcome in vitamin D RCTs, although immunomodulatory effects of vitamin D may well be relevant in the context of autoimmune or infectious diseases [107,108]. Notably, there is evidence suggesting that vitamin D supplementation may have an effect on lipid parameters and may modulate serum lipoprotein functions [71,82,83]. Moreover, we want to stress that insufficient evidence on beneficial effects of vitamin D on cardiovascular outcomes must not argue against indications for the prevention and treatment of vitamin D deficiency with reference to bone and overall musculoskeletal health [4,15,16,17,20].

Although there are still many vitamin D RCTs ongoing, we do not expect significantly altered results on vitamin D and CVD in the future, except in certain subgroups or in terms of very low effect sizes that are not highly relevant on an individual level, but may be of significance on a population level when considering public health issues. Moreover, given that there is much vitamin D RCT data available for post-hoc analyses, we have to be very cautious to avoid scientific misconduct with reference to “p-hacking” and HARKing (Hypothesizing After the Results are Known) that may cause misleading conclusions, in particular if combined with publication bias.

## 9. Conclusions

In summary, the available data do not support a major effect of vitamin D supplement use in the general population or vitamin D (metabolite) administration in the clinical setting to reduce CVD risk. However, more research is necessary to assess whether personalized preventive and therapeutic strategies are effective in some subgroups, i.e., individuals with a combination of low vitamin D status with specific gene variants and/or certain nutrition and lifestyle factors or those with severe vitamin D deficiency.

## Figures and Tables

**Figure 1 ijms-22-02896-f001:**
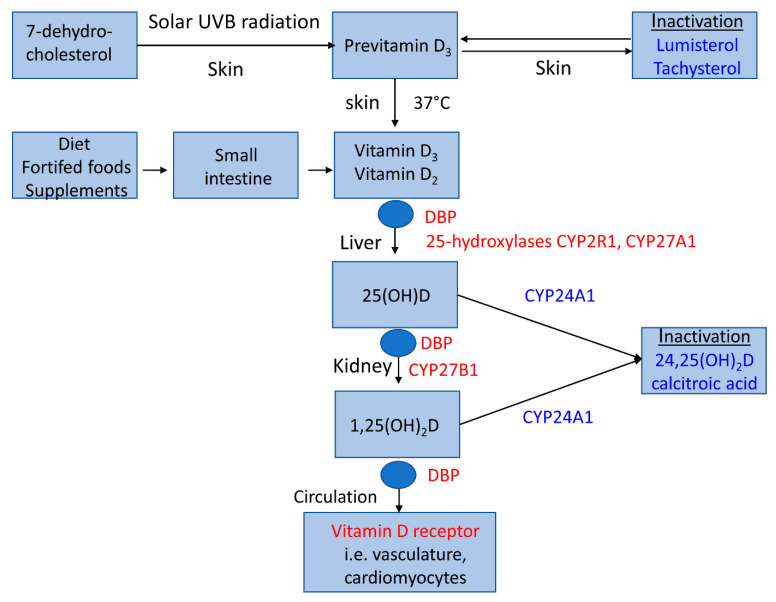
Major metabolic pathways of vitamin D in the human body. Previtamin D_3_ is produced in the skin by solar UVB radiation (290–315 nm) from 7-dehydrocholesterol and isomerized by a thermal reaction into vitamin D_3_ or reversely metabolized to vitamin D-inactive substances such as lumisterol and tachysterol. Vitamin D_3_ (cholecalciferol) or vitamin D_2_ (ergocalciferol) can also be ingested orally from native foods (fish, dairy, mushrooms), vitamin D-fortified foods, or supplements. Both cutaneously synthesized and orally ingested vitamin D are metabolized by hepatic 25-hydoxylases into 25-hydroxyvitamin D (25(OH)D), the main 25-hydroxylase being a microsomal enzyme (gene CYP2R1). A renal 1α-hydroxylation of 25(OH)D results in the formation of the physiologically active vitamin D hormone, 1,25-dihydroxyvitamin D (1,25(OH)_2_D). After release into circulation, 1,25(OH)_2_D can be taken up by various target tissues and exerts its effects by vitamin D receptor-mediated processes. Vitamin D binding protein (DBP) is the major transport protein for vitamin D and its metabolites. Inactivation of vitamin D is induced by 24-hydroxylase. Anabolic molecules are highlighted in red font, and catabolic molecules in blue font.

**Figure 2 ijms-22-02896-f002:**
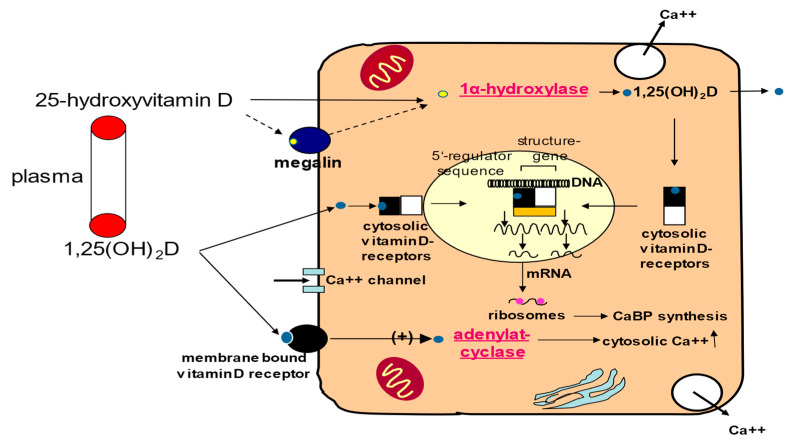
Proposed effects of vitamin D signaling in cardiomyocytes by rapid non-genomic and genomic effects via membrane-bound and cytosolic vitamin D receptors, respectively. Cytosolic binding of 1,25(OH)_2_D leads to hetero-dimerization of the vitamin D receptor (black rectangle) with the retinoid X receptor (white rectangle), and ultimately to translocation of this complex into the nucleus, where it binds to vitamin D response elements (orange rectangle). Additionally, 1,25(OH)_2_D may be produced locally by the internalization of 25(OH)D via diffusion or a vitamin D receptor (VDR)-independent, megalin–cubilin-mediated process (hypothetical). Non-genomic effects of 1,25(OH)_2_D result in the rapid release of ionized calcium from intracellular stores, whereas genomic effects result in calcium-binding protein (CaBP) synthesis. Locally produced 1,25(OH)_2_D may also exhibit paracrine actions. Yellow circle, 25(OH)D; blue circle, 1,25(OH)_2_D.

**Figure 3 ijms-22-02896-f003:**
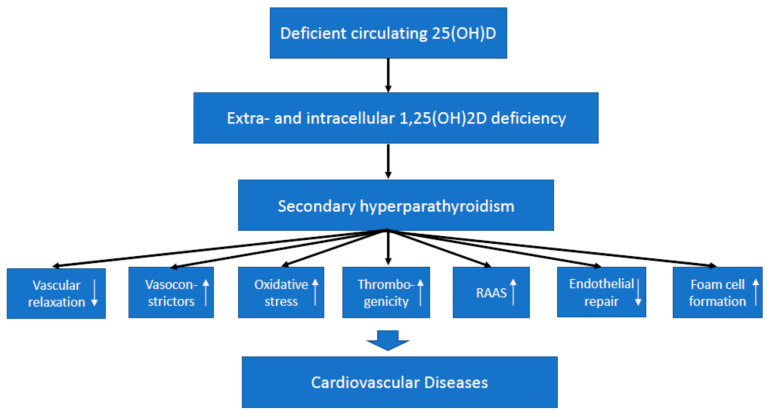
Suggested effect of vitamin D deficiency on the risk of cardiovascular disease. Notes: 25(OH)D, 25-hydroxyvitamin D; 1,25(OH)_2_D, 1,25-dihydroxyvitamin D; RAAS, renin–angiotensin–aldosterone system. White arrows indicate suppression or activation, where appropriate.

**Table 1 ijms-22-02896-t001:** Meta-analyses of observational studies on vitamin D status and cardiovascular events/outcomes.

Author, Year [Ref.]	Outcome	25(OH)D Reference Category	25(OH)D Category	Relative Risk (95%CI)
Wang, 2012 [47]	Cardiovascular disease	Highest category	Lowest category	1.52 (1.30–1.77)
Brodum-Jacobsen, 2013 [50]	Ischemic stroke	Highest quartile	Lowest quartile	1.54 (1.43–1.65)
Schöttker, 2014 [51]	CVD mortality without CVD history	Top quintile	Bottom quintile	1.41 (1.18–1.68)
	CVD mortality with CVD history	Top quintile	Bottom quintile	1.65 (1.22–2.22)
Chowdhury, 2014 [52]	CVD mortality	Top thirds	Bottom thirds	1.14 (1.01–1.29)
Zhang, 2017 [46]	CVD events	75 nmol/L	50 nmol/L	1.00 (0.95–1.05)
			25 nmol/L	1.08 (1.04–1.15)
	CVD mortality	75 nmol/L	50 nmol/L	1.04 (0.95–1.61)
			25 nmol/L	1.15 (1.05–1.22)
Qi, 2017 [53]	Hypertension	>75 nmol/L	74.99–50 nmol/L	1.09 (1.05–1.14)
			<50 nmol/L	1.24 (1.08–1.41)
Zhang, 2018 [54]	CVD mortality	-	Per 25 nmol/L decrease	1.41 (1.27–1.59)
Yang, 2019 [48]	CVD mortality	>75 nmol/L	25–50 nmol/L	1.16 (1.04–1.27)
			<25 nmol/L	1.47 (1.15–1.81)
Golami, 2019 [55]	CVD mortality	Highest category	Lowest category	1.54 (1.29–1.84)
Zhang, 2020 [49]	Hypertension	75 nmol/L	50 nmol/L	1.09 (1.04–1.17)
			25 nmol/L	1.30 (1.11–1.52)

25(OH)D, 25-hydroxyvitamin D; CVD, cardiovascular disease; -, no data provided; CI, confidence interval.

**Table 2 ijms-22-02896-t002:** Meta-analyses of randomized controlled trials on vitamin D supplementation and selected surrogate markers of cardiovascular disease.

Author, Year [Ref.]	Studies	Participants	Vitamin D Dose Equivalent	Duration	WMD or SMD (95%CI)
*Flow-mediated Dilation*					
Stojanović, 2015 [69]	8	529	1788 to 5000 IU/d; 2 µg paricalcitol/d	8 to 16 weeks	0.96% (−1.24% to 2.06%)
Dou, 2019 [72]	5	354	5042 IU/d; 1–2 µg paricalcitol/d	4 to 17 weeks	1.66% (−0.20% to 3.51%)
Pincombe, 2019, [74]	10	655	2000 to 7114 IU/d	4 to 24 weeks	1.17% (−0.20% to 2.54%)
Beveridge, 2018 [73]	12	785	900 to 5000 IU/d; 1–2 µg paricalcitol/d	4 to 52 weeks	0.49% (−0.13% to 1.11%)
*Pulse wave velocity*					
Rodríguez, 2016 [70]	10	827	1600 to 5700 IU/d	4 to 52 weeks	−0.10 m/s (−0.24 m/s to 0.04 m/s)
Mirhosseini, 2018 [71]	11	1019	600 to 4500 IU/d	4 to 52 weeks	−0.20 m/s (−0.46 m/s to 0.06 m/s)
Dou, 2019 [72]	3	235	2473 to 5042 IU/d; 2143 IU calcifediol plus 0.5 µg calcitriol/d	17 to 26 weeks	−0.93 m/s (− 1.71 m/s to −0.15 m/s)
Pincombe, 2019, [74]	21	2098	600 to 5714 IU/d; 2,143 IU calcifediol plus 0.5 µg calcitriol/d	8 to 57 weeks	−0.09 m/s (−0.24 m/s to 0.07 m/s)
Chen, 2020 [75]	10	918	1667 to 5000 IU/d	8 to 52 weeks	−0.29 m/s (−0.51 m/s to −0.06 m/s)
Beveridge, 2018 [73]	10	674	1000 to 5700 IU/d	5 to 52 weeks	0.04 m/s (−0.32 m/s to 0.41 m/s)
*Augmentation Index*					
Rodríguez, 2016 [70]	8	497	1667 to 4000 IU/d	4 to 52 weeks	−0.15% (−0.32% to 0.02%)
Mirhosseini, 2018 [71]	10	965	1000 to 4000 IU/d	5 to 52 weeks	−0.09% (−0.37% to 0.20%)
Pincombe, 2019, [74]	16	5117	600 to 7114 IU/d	4 to 52 weeks	0.05% (−0.10% to 0.19%)
Beveridge, 2018 [73]	14	1030	1000 to 5700 IU/d	4 to 52 weeks	0.00% (−1.30% to 1.30%)
*LVEF*					
Jiang, 2016 [76]	4	303	1000 to 7114 IU/d	12 to 39 weeks	4.11% (−0.91% to 9.12%)
Wang, 2019 [77]	5	422	3552 to 7114 IU/d	26 to 52 weeks	2.56% (−2.18% to 7.31%)
Zhao, 2018 [78]	7	538	1000 to 7114 IU/d	10 to 52 weeks	4.18% (0.36% to 7.99%)
*LVEDD*					
Zhao, 2018 [78]	6	499	1000 to 7114 IU/d	12 to 52 weeks	−2.31 mm (−4.15 mm to −0.47 mm)

IU, international unit; d, day; WMD, weighted mean difference; SMD, standardized mean difference; CI, confidence interval; LVEF, left ventricular ejection fraction; LVEDD, left ventricular end-diastolic diameter.

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
