# Peer review of "Vitamin D and Cardiovascular Disease: An Updated Narrative Review"

_ijms, 2021, doi:10.3390/ijms22062896_

Round 1

Reviewer 1 Report

The review by Zittermann and coworkers is an update focused on the role of Vitamin D in cardiovascular disesases. The review is extensive and detailed and quite well written, but I have some  comments.

Major comments

Lipid metabolism is very crucial in pathophysiology of cardiovascular disease, including atherosclerosis, thus this aspect cannot be excluded considering that this review is entitled “Vitamin D and Cardiovascular Disease: an updated narrative review”. A mention with respect to the role of VIT D in lipid metabolism would be worth to be added, besides the three references cited in the discussion.

In addition, the references cited are respect to vitamin D and quantitative lipid profile. However, emerging evidence suggests a crucial role of serum lipoprotein functions in CVD risk. In particular, the HDL function, measured as cholesterol efflux capacity (HDL CEC), seems to be a better predictor of CVD risk rather than HDL plasma levels. Some studies reported on the effect of Vitamin D supplementation on HDL CEC, an HDL aspect even more important than levels, that should be cited as well (e.g., Greco et al. Nutr Metab Cardiovasc Dis. 2018 Aug;28(8):822-829)

Minor comments

The chapter 5 is not linearly organized. It is not very clear to follow. Some diseases are mentioned in the subparagraph 5.1. Then, in the subsequent sub paragraphs the human studies have been addressed. I would better reorganize the different topics in a clearer subdivision, using more proper subtitles in order to help the reader to easily follow the various aspects.

I also would unbundle the paragraphs 5.4.3 and 5.4.4 from chapter 5.

Among the keywords, VDR should be spelled.

Finally, although the paper is very well written, it would be good make a language revision because I have detected some minor errors.

Author Response

Dear reviewer,

we thank you for the constructive comments! Our responses to your comments are listed below.

The review by Zittermann and coworkers is an update focused on the role of Vitamin D in cardiovascular disesases. The review is extensive and detailed and quite well written, but I have some  comments.

Answer: We thank the reviewer for this positive comment.

Major comments

Lipid metabolism is very crucial in pathophysiology of cardiovascular disease, including atherosclerosis, thus this aspect cannot be excluded considering that this review is entitled “Vitamin D and Cardiovascular Disease: an updated narrative review”. A mention with respect to the role of VIT D in lipid metabolism would be worth to be added, besides the three references cited in the discussion.

In addition, the references cited are respect to vitamin D and quantitative lipid profile. However, emerging evidence suggests a crucial role of serum lipoprotein functions in CVD risk. In particular, the HDL function, measured as cholesterol efflux capacity (HDL CEC), seems to be a better predictor of CVD risk rather than HDL plasma levels. Some studies reported on the effect of Vitamin D supplementation on HDL CEC, an HDL aspect even more important than levels, that should be cited as well (e.g., Greco et al. Nutr Metab Cardiovasc Dis. 2018 Aug;28(8):822-829)

Answer: We have now inserted a paragraph ‚Lipid parameters’. In this paragraph, we have also mentioned the effect of vitamin D on serum lipoproteins. The study by Greco et al. has been cited and discussed in the revised version of the manuscript. In detail we note that there is also evidence from a proof of principle study by Greco et al that vitamin D supplementation may modulate serum lipoprotein functions related to macrophage cholesterol homeostasis.

Minor comments

The chapter 5 is not linearly organized. It is not very clear to follow. Some diseases are mentioned in the subparagraph 5.1. Then, in the subsequent sub paragraphs the human studies have been addressed. I would better reorganize the different topics in a clearer subdivision, using more proper subtitles in order to help the reader to easily follow the various aspects.

I also would unbundle the paragraphs 5.4.3 and 5.4.4 from chapter 5.

Answer: The headings of chapter 5 have been replaced by more proper subtitles. Moreover, former paragraphs 5.4.3 and 5.4.4 are now paragraphs 6 and 7, respectively.

Among the keywords, VDR should be spelled.

Answer: Done. Many thanks!

Finally, although the paper is very well written, it would be good make a language revision because I have detected some minor errors.

Answer: The manuscript has now been checked with a native English speaker.

Reviewer 2 Report

General comment:

This manuscript, entitled “Vitamin D and Cardiovascular Disease: an updated narrative review,” authored by Zittermann et al., reports a brief survey on the role and implication of vitamin D in cardiovascular diseases. This review has an excellent introduction to vitamin D's molecular and genetic fate to regulate cell activity and how they get metabolized. This kind of review is rare to connect molecular events of vitamin D metabolism and its analog with cardiomyopathy's clinical aspect. In my opinion, this is a useful review and is suitable for publication in Int. J. Mol. Sci. after the authors have addressed the following comments and questions:

Specific comments:

  • In figure 1 where is vitamin D receptor (VDR) and vitamin D binding protein (DBP)stand as it is one of the first molecules to which vitamin D interact? The author can address distinguishing features like coloring for anabolic (VDR, CYP2R1, CYP27A1, CYP27B1) and catabolic (CYP24A1) events. The fate of catabolic molecules – calcitroic acid and lactones on cellular signaling events.
  • In figure 2 Megalin directly interact with vitamin D or via VDR? Role of Cubilin?
  • Autor can think of a figure where the endocrine system like PTH hormone regulates vitamin D to atherosclerosis to adverse cardiovascular events. That will justify Vitamin D and CVD connection.

Author Response

Dear reviewer,

we thank you for the constructive comments. Our responses to the comments are listed below.

This manuscript, entitled “Vitamin D and Cardiovascular Disease: an updated narrative review,” authored by Zittermann et al., reports a brief survey on the role and implication of vitamin D in cardiovascular diseases. This review has an excellent introduction to vitamin D's molecular and genetic fate to regulate cell activity and how they get metabolized. This kind of review is rare to connect molecular events of vitamin D metabolism and its analog with cardiomyopathy's clinical aspect. In my opinion, this is a useful review and is suitable for publication in Int. J. Mol. Sci. after the authors have addressed the following comments and questions:

 Answer: We thank the reviewer for this positive comment.

Specific comments:

In figure 1 where is vitamin D receptor (VDR) and vitamin D binding protein (DBP)stand as it is one of the first molecules to which vitamin D interact? The author can address distinguishing features like coloring for anabolic (VDR, CYP2R1, CYP27A1, CYP27B1) and catabolic (CYP24A1) events. The fate of catabolic molecules – calcitroic acid and lactones on cellular signaling events.

Answer: We thank the reviewer for this comment because we could improve our figure. In detail, DBP and VDR have been added to the figure. Anabolic and catabolic molecules have been highlighted in red and blue font, respectively. Calcitroic acid is now also included in the figure. We hope that the reviewer agrees with this modified figure.

In figure 2 Megalin directly interact with vitamin D or via VDR? Role of Cubilin?

Answer: It is now clarified that the process of internalization of DBP-bound 25(OH)D is mediated by a process which requires not only megalin, but also its co-receptor cubilin. There is some evidence that the megalin-cubilin transport of DBP-bound 25(OH)D does not require VDR (Negri. Nephrology 2006;11:510-5; Mason et al. Curr Dev Nutr. 2019;3:nzz087). This is now stated in the legend to Figure 2.  

Autor can think of a figure where the endocrine system like PTH hormone regulates vitamin D to atherosclerosis to adverse cardiovascular events. That will justify Vitamin D and CVD connection.

Answer: A figure (Figure 3) has been added, indicating the suggested effect of vitamin D deficiency on CVD risk mediated by secondary hyperparathyroidism. 

We have done all efforts to consider the reviewers’ comments and hope that the manuscript is now suitable for publication in IJMS.

Yours sincerely,

Armin Zittermann & Stefan Pilz

Round 2

Reviewer 1 Report

The authors addressed all the raised comments. The only comment I have is minor and it is reported below.

Given the new section on lipid parameters added in the review, a brief mention of what authors found from this section should be added also in the discussion.

Author Response

We thank the reviewer again for working on our manuscript.

According to this comment we now included the following sentence into the discussion: "Notably, there is evidence suggesting that vitamin D supplementation may have an effect on lipid parameters and may modulate serum lipoprotein functions [71, 82, 83]. "

We hope that the reviewer agrees with this.

Yours sincerely,

Armin Zittermann and Stefan Pilz